# Design and Preparation of Compact 3-Bit Reconfigurable RF MEMS Attenuators for Millimeter-Wave Bands

**DOI:** 10.3390/mi16101117

**Published:** 2025-09-29

**Authors:** Shilong Miao, Rui Chai, Yuheng Si, Yulong Zhang, Qiannan Wu, Mengwei Li

**Affiliations:** 1School of Semiconductors and Physics, North University of China, Taiyuan 030051, China; 19834791447@163.com; 2School of Instrument and Intelligent Future Technology, North University of China, Taiyuan 030051, China; s202306002@st.nuc.edu.cn (R.C.); 18834372826@163.com (Y.S.); zhangyl2024@nuc.edu.cn (Y.Z.); lmwprew@163.com (M.L.); 3Academy for Advanced Interdisciplinary Research, North University of China, Taiyuan 030051, China; 4Center for Microsystem Intergration, North University of China, Taiyuan 030051, China; 5School of Instrument and Electronics, North University of China, Taiyuan 030051, China; 6Key Laboratory of Dynamic Measurement Technology, North University of China, Taiyuan 030051, China

**Keywords:** T-type, RF MEMS, switch, reconfigurable step attenuator

## Abstract

As a core functional device in microwave systems, attenuators play a crucial role in key aspects such as signal power regulation, amplitude attenuation, and impedance matching. Addressing the pressing technical issues currently exposed by attenuators in practical applications, such as excessive insertion loss, low attenuation accuracy, large physical dimensions, and insufficient process reliability, this paper proposes a design scheme for an RF three-bit reconfigurable stepped attenuator based on radio frequency micro-electromechanical systems (RF MEMS) switches. The attenuator employs planar integration of the T-type attenuation network, Coplanar Waveguide (CPW), Y-shaped power divider, and RF MEMS switches. While ensuring rational power distribution and stable attenuation performance over the full bandwidth, it reduces the number of switches to suppress parasitic parameters, thereby enhancing process feasibility. Test results confirm that this device demonstrates significant advancements in attenuation accuracy, achieving a precision of 1.18 dB across the 0–25 dB operational range from DC to 20 GHz, with insertion loss kept below 1.65 dB and return loss exceeding 12.15 dB. Additionally, the device boasts a compact size of merely 0.66 mm × 1.38 mm × 0.32 mm, significantly smaller than analogous products documented in existing literature. Meanwhile, its service life approaches 5 × 10^7^ cycles. Together, these two attributes validate the device’s performance reliability and miniaturization advantages.

## 1. Introduction

RF MEMS attenuators have become the core passive components of high-frequency systems, such as 5G/6G communications, satellite payloads, and phased-array radars, by virtue of their low insertion loss, high linearity, and miniaturization [1,2,3]. By dynamically adjusting signal attenuation, the device significantly extends the dynamic range in microwave test systems. It suppresses system saturation from strong signals while enhancing weak signal detection. Additionally, it protects front-end equipment with a blocking failure protection mechanism and optimizes impedance matching to reduce signal reflection and improve transmission efficiency. Furthermore, the attenuator effectively minimizes signal coupling and electromagnetic interference between multiple channels, thereby enhancing the testing accuracy of spectrum analyzers, vector network analyzers, and signal sources [4,5,6,7]. In practical applications, RF MEMS attenuators have been deeply integrated into military and civilian high-frequency systems. In the military field, they are widely used in radar systems [8], satellite communications [9], and multi-band communication networks [10,11,12] to achieve reliable operation in complex electromagnetic environments by precisely controlling the signal power; in the civil field, their miniaturization and high-performance characteristics provide key support for 5G base stations, satellite Internet terminals and high-speed data links.

In contemporary RF/microwave systems, attenuators play a crucial role in signal power management, significantly influencing dynamic range, linearity, and stability. Traditional resistive attenuators offer simple structures and low costs but suffer from significant parasitic effects at high frequencies, reducing attenuation accuracy. PN-junction diode attenuators achieve continuous attenuation via bias voltage modulation of junction resistance, yet their limited power capacity and temperature sensitivity limit use in complex settings. FET digital attenuators facilitate rapid programmable control but face challenges related to conduction resistance and parasitic capacitance that impede their performance in wideband, high-precision applications. As communication technology advances toward higher frequencies, miniaturization, and intelligent integration, the drawbacks of traditional attenuators are more evident. MEMS-based switches offer a promising solution, using microfabrication to deeply integrate mechanical and electrical functions in attenuators. This yields devices with low insertion loss, high isolation, ultra-compactness, and outstanding linearity when paired with precision attenuation networks.

So far, researchers in the microwave field have carried out a great deal of research on attenuators. In 2017, Duy P. Nguyen et al. designed a monolithic microwave integrated circuit (MMIC) attenuator based on gallium arsenide FET technology, which for the first time employed two-dimensional stacked FETs in its structure. The results indicated that this step attenuator operated over a wide frequency range of 1.5~45 GHz, with an insertion loss ranging from 1.9 to 5.5 dB and a maximum attenuation of 26 dB [13]. In 2021, Clifford D. Cheon et al. developed a DC~67 GHz step attenuator utilizing silicon–germanium bipolar complementary metal-oxide-semiconductor technology. This attenuator achieved an attenuation range of 31.5 dB with a step size of 0.5 dB and maintained an insertion loss below 7.8 dB [14]. In 2022, Kim et al. designed a four-bit digital step attenuator operating within the frequency range of 0.5 to 12 GHz, achieving an attenuation coverage of up to 30 dB with an accuracy of 2 dB; the insertion loss varied between 2.8 and 8.3 dB while ensuring return loss was less than 12 dB [15]. In addition, in 2024 Haipeng Fu et al. demonstrated a broadband high-linearity seven-bit digital step attenuator capable of delivering a remarkable attenuation range of up to 31.7 dB across the DC to 7 GHz frequency band; it featured minimum step accuracy at 0.25 dB and overall attenuation accuracy within 0.5 dB alongside an insertion loss better than or equal to 2.7 dB [16]. At present, notable research gaps persist in RF MEMS attenuators. Key performance indicators like insertion loss, attenuation accuracy, return loss, and frequency coverage still fall short of the ideal. Balancing these performance aspects proves challenging, rendering simultaneous optimization unattainable. Further investigation is needed to optimize their microwave characteristics [17,18,19,20,21].

In this paper, a 3-bit MEMS digital attenuator is proposed. By combining the differentiated T-type attenuation network with the RF MEMS switches in the CPW, the attenuator not only reduces the process complexity, but also realizes the compact design of the device structure, which effectively improves the reliability and engineering realizability through the optimization of the number of switches under the premise of guaranteeing the function realization. In terms of performance, the attenuator can precisely regulate the reconfigurable attenuation value over a wide frequency range of DC-20 GHz, which significantly improves the attenuation accuracy. Meanwhile, through the optimization of impedance matching, the attenuator provides a solution with both high precision and flexibility for signal attenuation, and the attenuation can be dynamically adjusted according to the signal strength of the actual application, which can meet the needs of diversified systems.

## 2. 3-Bit RF MEMS Attenuator Design

### 2.1. Topological Construction Design of Attenuator

Figure 1 shows the schematic diagram of the attenuator, the input port and output port are cascaded by three-stage attenuation bits, each of which consists of a T-type attenuation network, two Y-type power dividers and two RF MEMS switches. Through the switch on/off combination, the signal path can be dynamically selected: when the switch is set to the attenuation network, the signal undergoes a predetermined attenuation via a T-type network. Conversely, when the switch connects to the direct circuit, the signal is transmitted with minimal loss. In this paper, the CPW is used as the basic transmission line structure to realize the precise distribution of power absorption attenuation by controlling the on–off states of six RF MEMS switches and dynamically switching different combinations of T-type attenuation networks. This design adopts a synergistic architecture that integrates a Y-shaped power divider with a T-shaped attenuation network. This innovative integration significantly reduces the number of required MEMS switches. This not only shortens the effective electrical length and simplifies the attenuator’s structure for a compact design, but also mitigates the risk of “device-wide failure due to a single switch malfunction” by minimizing switch count.

Here, the 5 dB, 10 dB, and 10 dB attenuation modules are sequentially defined as the first, second, and third bit. By configuring the switch channels, the signal path can be controlled to either pass through the attenuation network or travel via the direct-through line. When a switch is on, the state is defined as “1”; when a switch is off, the state is set to “0”. Based on the control of each attenuation module by the switches of this 3-bit attenuator, attenuation adjustment within the range of 0 to 25 dB with a step of 5 dB can be achieved. The relationship between each state and the corresponding attenuation value is detailed in Table 1.

### 2.2. Attenuation Network Module Design

Since T-type attenuation networks are particularly well-suited for applications that require lower-level attenuation—typically ranging from 1 to 10 dB—they offer several advantages, including minimal parasitic effects and stable performance in high-frequency environments. This paper employs a T-type TaN resistor network with a symmetric topology as the attenuation module. A systematic evaluation of its high-precision attenuation characteristics is conducted. The attenuation module consists of two series resistors (R_Sh_) positioned at the center of the signal line, accompanied by a shunt resistor (R_Se_), as illustrated in Figure 2a.

The equivalent circuit representation of this configuration is depicted in Figure 2b. The relationship between R_Sh_, R_Se_, and the magnitude of attenuation (A_dB_) is defined by Equations (1) and (2), which are presented as follows:(1)$$R_{Sh}=Z_{0}\cdot \left(\frac{10^{\frac{A_{dB}}{20}}-1}{10^{\frac{A_{dB}}{20}}+1}\right)$$(2)$$R_{Se}=2Z_{0}\cdot \left(\frac{10^{\frac{A_{dB}}{20}}}{10^{\frac{A_{dB}}{20}}-1}\right)$$

In the equation, Z_0_ (typically 50 Ω) represents the port characteristic impedance, while A denotes the attenuation level (dB). Using Equations (1) and (2), we calculate the resistor values for 5 dB (R_Sh_ = 14.0 Ω, R_Se_ = 82.2 Ω) and 10 dB (R_Sh_ = 26.0 Ω, R_Se_ = 35.2 Ω) attenuator networks. These values are derived under ideal transmission line cascading conditions; practical implementation requires Ansys-based optimization for impedance matching.(3)$$R=R_{e}\times \frac{L}{W}$$

Here, L represents the length of the resistor, W denotes the width of the resistor, and R_e_ indicates the square resistance value of the thin film. Consequently, we derive the corresponding resistor dimensions for each attenuation. Subsequently, we calculate these dimensions using Equation (3), and optimize the thin-film resistance for all attenuator modules through Ansys EM 2023 simulation software. For every resistor module within the network, the thin film is specified to have a square resistance value of 157 Ω/□. The final structural parameters are summarized in Table 2.

### 2.3. RF MEMS Switch Desigin

RF MEMS switches are pivotal components that facilitate the compact, miniaturized integration of reconfigurable attenuators [22,23]. In this paper, we employ a RF MEMS switch featuring a straight cantilever beam design, and integrate it into a 3-bit step attenuator. This switch primarily consists of a cantilever beam, contact pads, anchors, and release holes, with its detailed structural configuration illustrated in Figure 3a,b. The release holes seamlessly incorporated into the upper electrode markedly accelerate the release procedure of the cantilever beam, thereby assuming a pivotal role in enhancing the manufacturing yield of the cantilever beam structure and promoting its extensive, large-scale production. The switch demonstrates excellent performance in the frequency range of DC-25 GHz, with an insertion loss better than 0.1 dB and an isolation better than 20 dB.

The main geometric parameters of the proposed switch are summarized in Table 3.

To further verify the operational reliability of the switch, this study employed COMSOL Multiphysics 5.6 to carry out finite element simulation analysis. Given the short dimensions of the designed cantilever beam, it exhibits high structural rigidity, with a calculated critical stress value of 200 MPa. Figure 4a displays the stress simulation diagram of the cantilever beam, indicating that the root area where the cantilever beam connects to the fixed end is the primary region of stress concentration: when a driving voltage is applied, this part bears the dominant bending load. In the simulation diagram, this area is marked in red to denote the stress peak, with the corresponding simulated peak stress being only 15.4 MPa, which is below the material’s fracture strength threshold. This confirms that the switch poses no risk of structural failure under operational loads.

Figure 4b illustrates the COMSOL-simulated downward deflection state of the cantilever beam under the action of the pull-in voltage (V_pull-in_), which not only provides an intuitive verification of the stability of the switch’s operational performance but also confirms that it can stably achieve a downward displacement of 2 μm, highly consistent with the design objective. Additionally, Figure 4c presents the relationship curve between V_pull-in_ and the displacement of the cantilever beam. The results show that when the displacement approaches 2/3 Gap (1.35 μm), V_pull-in_ reaches a maximum value of 38.9 V. This maximum value represents the V_pull-in_ of the switch, offering crucial parameter references for the subsequent design of the circuit driver.

### 2.4. Frequency Response Flattening Design of Attenuator Compensation Mechanism

As illustrated in Figure 1, leveraging microwave transmission theory and advanced circuit topology optimization, the T-network attenuator integrated with a Y-shaped power divider architecture demonstrates significant performance advantages over conventional Single-Pole Double-Throw (SPDT) switch-based MEMS attenuators, particularly in impedance matching and attenuation precision. Traditional SPDT configurations necessitate a minimum of four switches to enable single-stage attenuation and through-path operation, leading to complex switch matrices and extended transmission lines. These significantly increase parasitic capacitance and inductance. By contrast, the proposed design employs a Y-shaped power divider to halve both switch count and transmission line length, effectively minimizing electrical length (the phase delay along the signal propagation path) and mitigating parasitic effects at their origin.

From microwave transmission theory, electrical length (physical length/wavelength) determines signal phase and impedance transformation. In conventional attenuators, extended transmission lines and multi-stage switching increase electrical length. This introduces parasitic elements, which degrade impedance matching (resulting in higher return loss) and distort voltage division ratios. As a result, attenuation accuracy is reduced. The proposed design uses Y-shaped divider symmetry to balance signal power, ensuring a phase deviation of less than 5° between paths. Its T-network layout limits the electrical length of critical segments to less than 0.05 wavelengths (1.5 mm), significantly minimizing reflections.

We, respectively, incorporated the CPW, air bridge, Y-shaped power divider, and RF MEMS switch into the 5 dB and 10 dB attenuation networks, and then constructed design models for the 5 dB and 10 dB attenuation units to verify their attenuation flatness characteristics. The simulation results demonstrate that the two single-bit attenuators exhibit exceptional attenuation performance. Specifically, as illustrated in Figure 5, the simulated attenuation values (S21) for the 5 dB and 10 dB attenuation modules are recorded at 5.32 ± 0.17 dB and 10.22 ± 0.27 dB, respectively, with both achieving an attenuation accuracy within 0.27 dB. Throughout the entire operational frequency range (up to a maximum of 20 GHz), their matching performance exceeds 20.02 dB. Based on these performance validations, both sets of attenuators conform to the design specifications for programmable attenuation and are suitable for cascading systems. To achieve a reconfigurable attenuation function ranging from 0 to 25 dB in increments of 5 dB, this study employs a cascaded network architecture comprising three bits of attenuators to meet the design requirements.

## 3. Simulation

As shown in Figure 6, a reconfigurable attenuator is designed by integrating resistor attenuation modules, RF MEMS switches, transmission lines, and power dividers. This attenuator consists of three attenuation bits (5 dB, 10 dB, and 10 dB) and is equipped with six RF MEMS switches to enable attenuation adjustment from 0 to 25 dB in steps of 5 dB. In the simulated structure, BF33 glass is selected as the substrate material owing to its low loss tangent (4.9 × 10^−6^) and dielectric constant (4.6); gold (Au) is chosen for the air bridges, upper electrodes, and CPW components on account of its chemical stability and ease of fabrication in semiconductor processes. The resistor material is a 30 nm thick TaN thin film with a sheet resistance of 157 Ω/□. When the switches in the DC paths below each attenuation bit are turned on, the attenuator operates in the through mode (i.e., signal transmission without attenuation); conversely, when the switches in the attenuation network are activated, the attenuator enters the attenuation mode (i.e., signal attenuation according to the preset ratio). Table 4 lists the main design parameters of this device.

Systematic electromagnetic simulations were conducted using Ansys EM 2023 simulation software to evaluate the microwave performance of the attenuator. The results are presented in Figure 7. Simulation data for the 3-bit MEMS attenuator demonstrate that across the broad frequency range from DC to 20 GHz, key RF metrics such as attenuation accuracy and return loss remain within ideal ranges, fully validating its excellent RF performance and wideband applicability. The insertion loss of the attenuator is within 1.52 dB (0 dB mode). All attenuation states maintain an accuracy within 0.8 dB. Matching networks for all six attenuation states exhibit exceptional performance, with return loss exceeding 13.16 dB (VSWR < 1.56). The 15 dB state achieves the highest attenuation accuracy (<0.44 dB). The 5 dB state demonstrates the best return loss (>22.72 dB).

Compared to previous studies, this attenuator integrates a T-type TaN resistor network with MEMS switches, thereby achieving enhanced attenuation accuracy and a reduced device size while maintaining broadband performance and effective impedance matching. A summary of the detailed results is presented in Table 5

Given the multi-mode operational features of the attenuator, this study visually represents the optimized performance of a 3-bit attenuator by selecting three attenuation states: 0 dB, 15 dB, and 25 dB. Current density distribution maps (shown in Figure 8) illustrate the electromagnetic characteristics and performance advantages across different operating modes. A gradient color scale differentiates current density levels within signal line areas. When the attenuation module is engaged, there is a significant reduction in current density, indicating that the attenuation network effectively degrades signals by absorbing electromagnetic wave energy, thus ensuring stable attenuation accuracy. In contrast, when the direct current path is activated, current density shows a uniform distribution; this indicates lossless signal transmission throughout the direct current pathway structure. This uniformity minimizes insertion loss while preserving signal integrity and suppressing hotspot formation as well as local overheating phenomena. These factors contribute to enhanced device reliability and long-term operational stability.

## 4. Processing and Manufacturing

The 3-bit MEMS attenuator is manufactured using micro-nano surface technology, and the manufacturing process is schematically illustrated in Figure 9, which includes seven steps as described below.

(a)A 400 nm thick silicon nitride (Si_4_N_3_) film was prepared using plasma-enhanced chemical vapor deposition (PECVD) technology, followed by reactive ion etching (RIE) to etch the film into a bump structure pattern, as shown in Figure 9a.(b)A 30 nm thick TaN film was deposited using magnetron sputtering as the resistive layer material. A metal dry etching process was used to prepare a TaN attenuation network on a glass substrate. The TaN resistive structure is shown in Figure 9b.(c)An ammonia–hydrogen peroxide mixed solution was used to etch the 640 nm thick aluminum (Al) layer deposited by sputtering to form the pull-down electrode structure. Concurrently, a 400 nm thick Si_4_N_3_ film was patterned using RIE to serve as a dielectric isolation layer between the drive electrode and the upper electrode, as well as a protective coating for the drive circuit, as shown in Figure 9c.(d)A titanium (Ti)/Au bilayer film, with a total thickness of 200 nm, was sputter-deposited to serve as a seed layer. This bilayer comprises a 50 nm thick titanium layer and a 150 nm thick gold layer. An electroplating mold was subsequently prepared using a 6.7-μm AZ4620 photoresist through a photolithography process. Following this, a 1.28-μm gold layer was electroplated to create the CPW structure. The excess seed layer was then eliminated via an ion beam etching (IBE) dry etching process, yielding the final CPW structure as depicted in Figure 9d.(e)Polyimide is uniformly spin-coated onto the wafer. Following a pre-curing process and subsequent curing at elevated temperatures, A 2-μm thin film is formed to act as the sacrificial layer for the cantilever beam. The anchor point area of the cantilever beam is then etched using RIE during the photolithography process, as illustrated in Figure 9e.(f)A 150 nm Au layer was sputter-deposited onto the cured polyimide wafer as a seed layer. An electroplating mold was then prepared using a 6.7-μm AZ4620 photoresist through photolithography. Next, a 1.28-μm gold layer was electroplated to form the upper electrode of the cantilever beam. The remaining seed layer was removed via IBE, resulting in the cantilever beam structure shown in Figure 9f.(g)Oxygen plasma dry etching is utilized to precisely remove the polyimide sacrificial layer, enabling the release of the device. Given that this process operates under high-temperature conditions, it is crucial to control the release time per cycle to avoid excessive duration, which could otherwise lead to warping of the top electrode. To ensure the complete elimination of the sacrificial layer, multiple cycles are necessary, as demonstrated in Figure 9g.

**Figure 9 micromachines-16-01117-f009:**
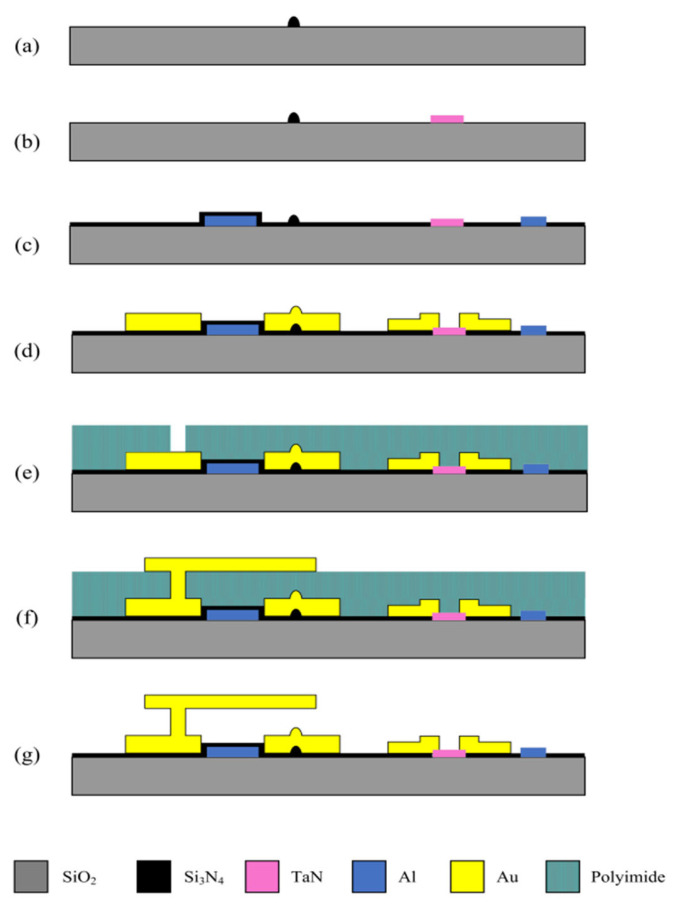
Device Process Flow Chart: (**a**) Fabricate contacts; (**b**) Fabricate resistor networks; (**c**) Fabricate driving electrodes; (**d**) Fabricate CPW; (**e**) Fabricate anchor points; (**f**) Fabricate cantilevers; (**g**) Release sacrificial layers.

The physical sample is depicted in Figure 10, showcasing a successfully fabricated 3-bit reconfigurable attenuator. In Figure 10a, the overall structure of the 3-bit attenuator is characterized using high-resolution scanning electron microscopy (SEM), which clearly reveals the microscopic morphology of the device along with the integrated layout of its functional modules. Figure 10b–d provide detailed illustrations of the local structures pertaining to the MEMS switch, T-type attenuator network, and Y-type power distributor, each exhibiting distinct geometric contours and excellent processing accuracy. This validates the feasibility of the proposed design scheme.

## 5. Testing Results

### 5.1. RF Characterization

This study utilizes the PNA-XN5242B vector network analyzer and employs the SOLT (Short-Open-Load-Thru) two-port calibration method to assess the RF performance of the 3-bit MEMS attenuator described herein. The measured characteristics are presented in Figure 11. The results demonstrate that across the frequency range from DC to 20 GHz, the insertion loss of the attenuator is maintained at less than 1.65 dB, with a maximum attenuation deviation controlled within 1.18 dB, and a return loss exceeding 12.15 dB. The test results demonstrate that the 3-bit attenuator, which is based on MEMS technology, achieves a coordinated optimization of low loss, high precision, and excellent matching over a wide frequency range. This characteristic renders it particularly suitable for applications with stringent requirements concerning the dynamic range of RF front ends, such as those found in 5G millimeter-wave base stations and phased array radars.

Overall, the measurement results exhibit a strong alignment with the simulation outcomes, with comprehensive data presented in Table 6. Nevertheless, when juxtaposed with the simulation results, the fabricated devices inferior return loss performance and discrepancies in attenuation values. This variance can be traced back to two distinct categories of uncertainty errors stemming from the manufacturing process.

During the production phase, the actual resistance values of resistor networks may stray from the pre-defined design specifications. For example, subpar photolithography or etching precision could induce dimensional variations in resistor strips. Furthermore, throughout the wafer fabrication procedure, impurities or oxide layers might accumulate on the device surfaces. This, in turn, leads to an elevation in contact resistance and a degradation of matching characteristics. Collectively, these factors underpin the inaccuracy observed in the measurement results pertaining to return loss and attenuation values.

Table 7 presents a performance comparison between the proposed attenuator and other digitally controlled attenuators. The MEMS attenuator proposed in this paper combines MEMS switches with Y-type power dividers and T-type attenuation networks, which not only achieves superior overall RF performance and a smaller device size but also enables key indicators such as insertion loss, return loss, and attenuation accuracy to reach a more balanced level. This design provides an effective technical approach for the performance optimization and miniaturization of MEMS attenuators, thus holding value for further in-depth research.

### 5.2. Mechanical Characteristics

Furthermore, the mechanical performance of RF MEMS switches significantly influences the reliability of attenuators. In this study, 10 randomly selected attenuator samples were tested, and the results indicated that the average driving voltage range of their MEMS switches was 59 V, with a closing time of 58.5 μs and an opening time of 18.6 μs. Notably, the pull-in voltage during the simulation phase was 38.9 V, which differed significantly from the value observed in the test phase. This discrepancy is primarily attributed to residual stress, which induces deformation at the contact end of the cantilever beam and thereby widens the electrode gap.

This paper explores the key factors affecting the reliability of RF MEMS switches. In this domain, switch reliability is typically characterized by the number of operating cycles prior to failure. To promote stable contact formation and improve the switching speed of RF MEMS switches, the actual driving voltage in practical applications must exceed the pull-in voltage; industrially, this voltage is commonly set to 1.2–1.5 times the pull-in voltage. Guided by this principle, the signal generator in this study was configured with a driving signal amplitude of 90 V and a frequency of 10 kHz. As illustrated in Figure 12, the RF MEMS switch maintains a contact resistance consistently below 0.65 Ω when the number of operating cycles is within 5 × 10^7^, fully satisfying the performance requirements of RF attenuators.

## 6. Conclusions

This paper presents a design scheme for a 3-bit reconfigurable step attenuator based on fundamental unit modules. The core module comprises a Y-shaped power divider, RF MEMS switches, and a T-type resistive attenuation network. By cascading three basic units, this attenuator achieves an attenuation range of 0 to 25 dB in increments of 5 dB across a broad frequency spectrum from DC to 20 GHz. Experimental results demonstrate that the device exhibits an insertion loss of less than 1.65 dB throughout the entire frequency band, with an attenuation accuracy better than 1.18 dB. Furthermore, the 3-bit MEMS attenuator not only features a compact size (0.66 mm × 1.38 mm × 0.32 mm) but also has half the number of switches compared to traditional 3-bit MEMS attenuators, effectively preventing overall failure caused by the malfunction of a single switch. This design integrates high integration density with attenuator reliability while ensuring excellent comprehensive RF performance, demonstrating significant application potential in electronic communication and satellite radar systems.

## Figures and Tables

**Figure 1 micromachines-16-01117-f001:**
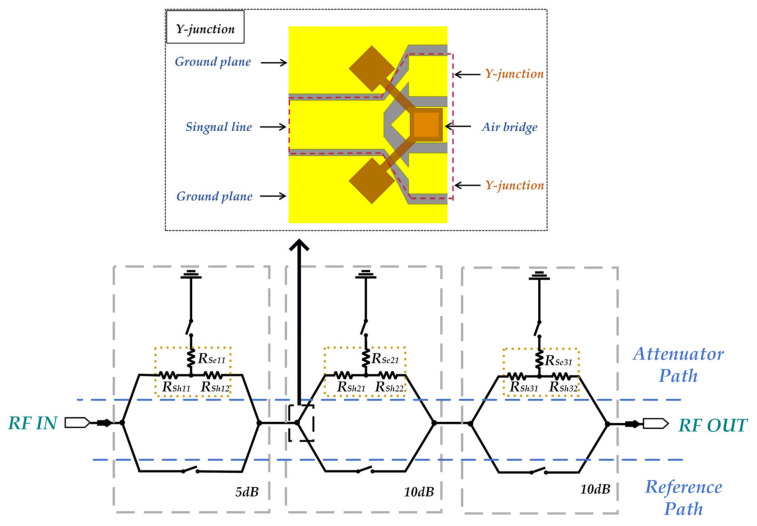
Schematic diagram of an attenuator.

**Figure 2 micromachines-16-01117-f002:**
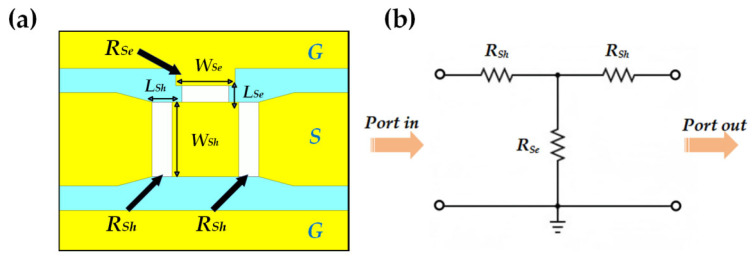
Structure of T-type resistor network: (**a**) Top view of T-type attenuation resistor in Ansys; (**b**) Equivalent diagram of T-type resistor.

**Figure 3 micromachines-16-01117-f003:**
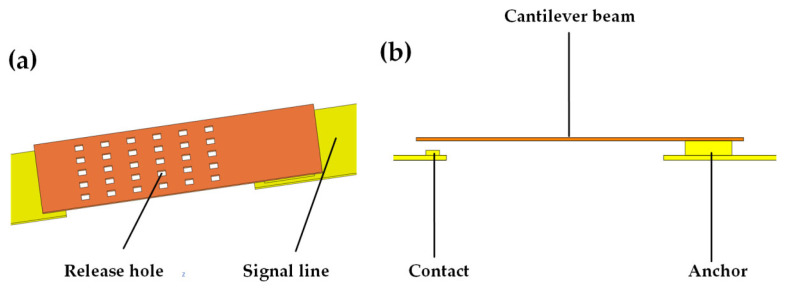
The Structure of the MEMS Switch: (**a**) Schematic 3D View; (**b**) Cross-Sectional View.

**Figure 4 micromachines-16-01117-f004:**
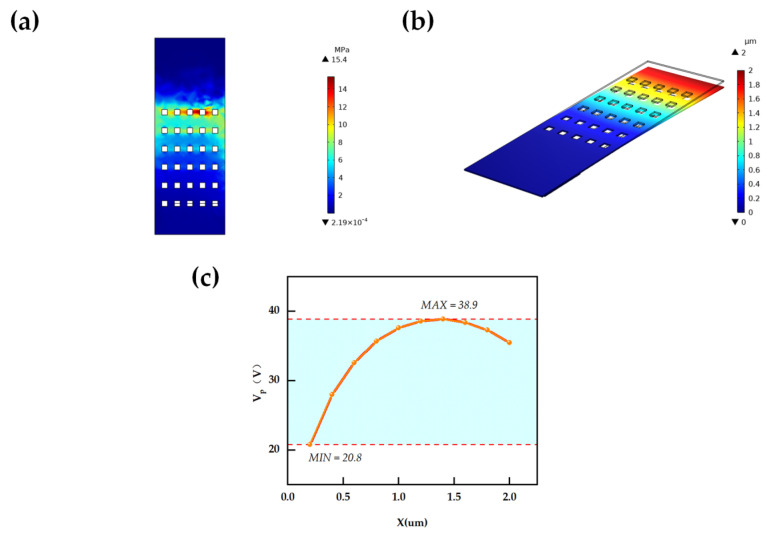
Mechanical Performance Analysis of MEMS Switch: (**a**) Stress Simulation of Cantilever Beam; (**b**) Simulation of Cantilever Beam in Pulled-Down State; (**c**) V_pull-in_ Corresponding to Different Displacements of Cantilever Beam.

**Figure 5 micromachines-16-01117-f005:**
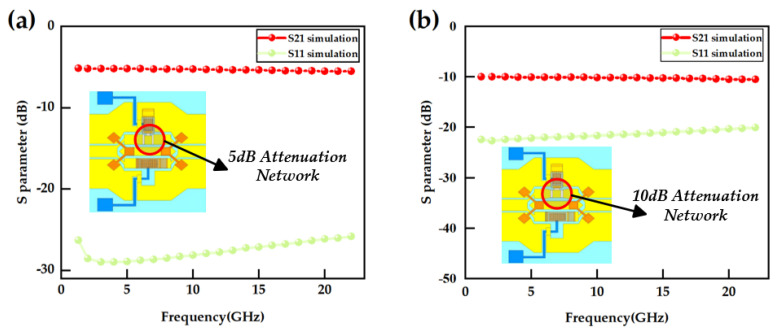
RF Simulation Performance Results: (**a**) 5 dB model; (**b**) 10 dB model.

**Figure 6 micromachines-16-01117-f006:**
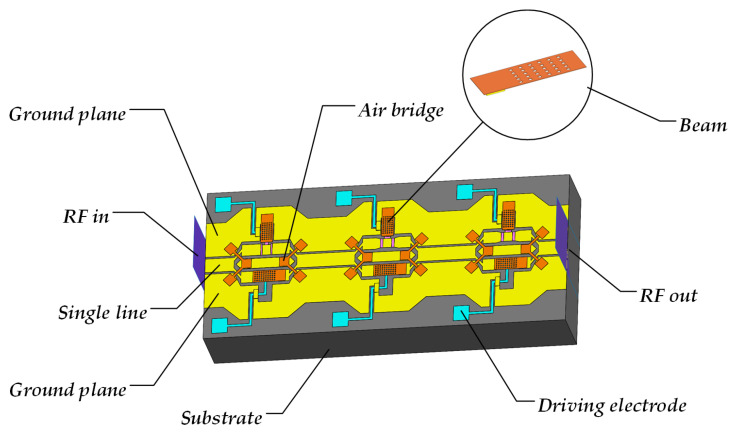
3-bit Reconfigurable MEMS Attenuator Ansys Model.

**Figure 7 micromachines-16-01117-f007:**
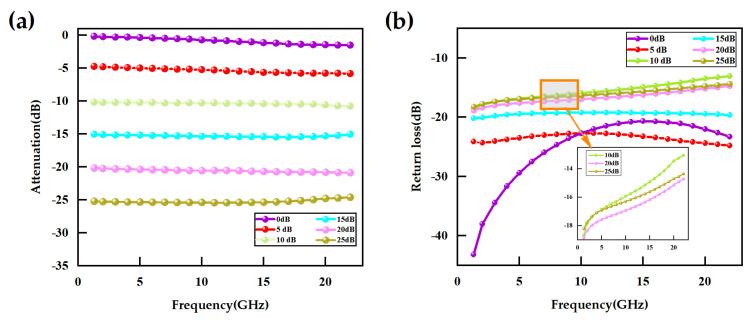
Simulation results of the attenuator: (**a**) Attenuation; (**b**) Return loss.

**Figure 8 micromachines-16-01117-f008:**
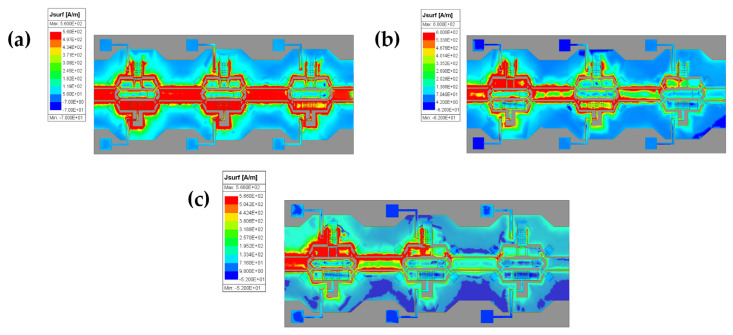
Current density distribution patterns of the 3-bit attenuator under different operating modes: (**a**) 0 dB attenuation state; (**b**) 15 dB attenuation state; (**c**) 25 dB attenuation state.

**Figure 10 micromachines-16-01117-f010:**
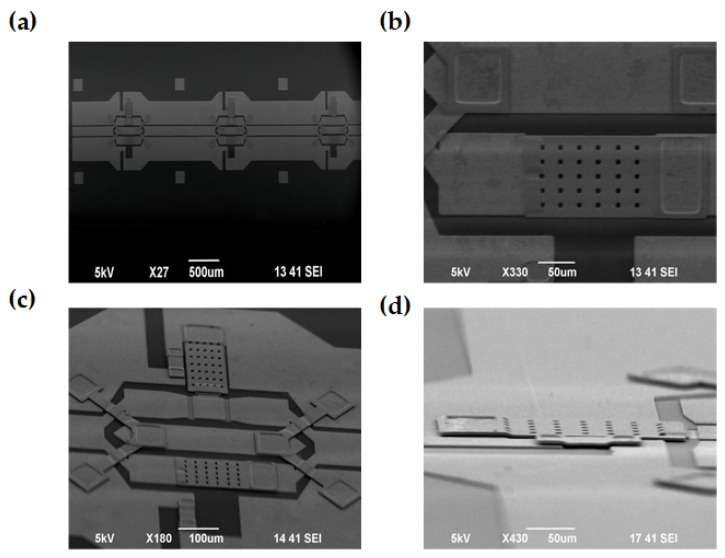
SEM images: (**a**) Top view of the 3-bit attenuator; (**b**) Top view of the cantilever beam; (**c**) 60° cross-sectional view of the attenuation network; (**d**) 90° cross-sectional view of the cantilever beam.

**Figure 11 micromachines-16-01117-f011:**
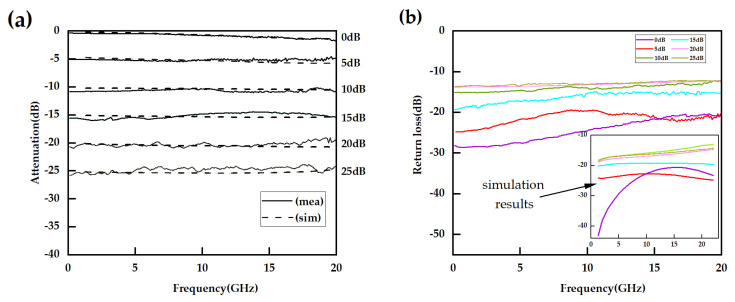
Simulated and measured RF characteristics for different states: (**a**) Attenuation value; (**b**) return loss.

**Figure 12 micromachines-16-01117-f012:**
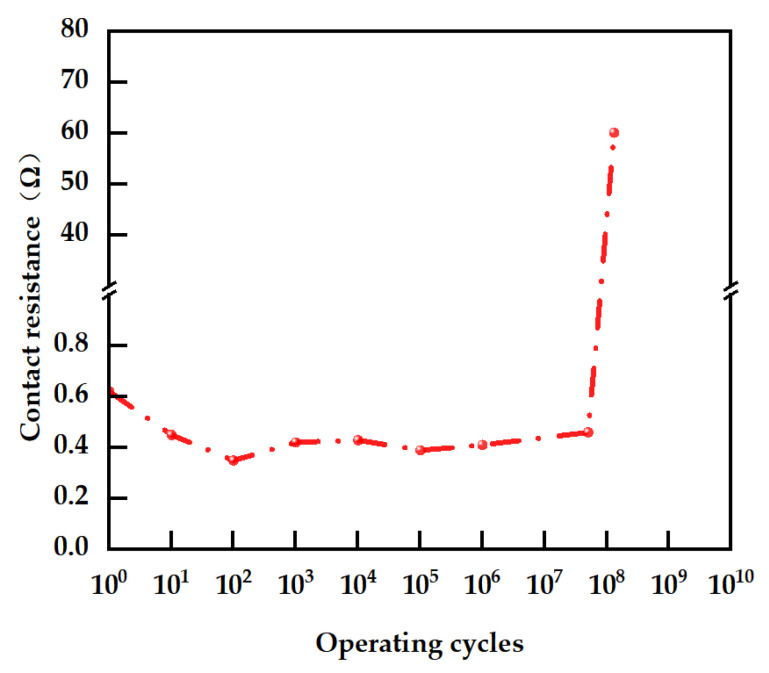
RF MEMS Switch Lifetime Test Curve.

**Table 1 micromachines-16-01117-t001:** The corresponding relationship between the target attenuation of the attenuator and the switch states.

Attenuation	The 1st Bit		The 2nd Bit		The 3rd Bit	
	0 dB	5 dB	0 dB	10 dB	0 dB	10 dB
0 dB	1	0	1	0	1	0
5 dB	0	1	1	0	1	0
10 dB	1	0	0	1	1	0
15 dB	0	1	0	1	1	0
20 dB	1	0	0	1	0	1
25 dB	0	1	0	1	0	1

**Table 2 micromachines-16-01117-t002:** Optimized Resistance Values and Dimensions of the Resistive Network.

A/dB		R_Sh_	R_Se_
5.0	Design value (Ω)	15.1	96.5
	L/W (µm/µm)	4.8/51.2	9.5/15.3
10.0	Design value (Ω)	28.5	39.3
	L/W (µm/µm)	7.5/41	6.4/25.6

**Table 3 micromachines-16-01117-t003:** The main geometric parameters.

Parameters	Values (µm)
Cantilever width	50
Cantilever length	138
Cantilever thickness	1.28
Gap	2

**Table 4 micromachines-16-01117-t004:** Main Structural Dimension Parameters of the 3-bit Attenuator.

Parameters	Values (µm)
Single line width	60
Ground plane width	148
CPW gap	8
CPW thickness	1.28
Driving electrode Size	70 × 35
Substrate size	660 × 1380 × 320

**Table 5 micromachines-16-01117-t005:** Simulated RF Performance of the 3-bit Attenuator.

State	Attenuation (dB)			S11 (dB/@20 GHz)
	Average ^1^	Accuracy	Error ^2^	
0 dB	0.76	1.52	0.76	21.12
5 dB	5.25	0.76	0.25	22.72
10 dB	10.49	0.79	0.49	13.16
15 dB	15.25	0.44	0.25	19.24
20 dB	20.47	0.72	0.47	14.72
25 dB	24.95	0.48	0.05	14.36

^1^ The average of the maximum and minimum attenuation across the entire operating frequency band. ^2^ The difference between the average value across the entire frequency band and the preset attenuation value.

**Table 6 micromachines-16-01117-t006:** The RF performance measurement results of the 3-bit attenuator.

State	Accuracy (dB)		S11 (dB/@20 GHz)	
	Simulation	Measurement	Simulation	Measurement
0 dB	1.45	1.65	21.12	20.34
5 dB	0.76	0.65	22.72	19.40
10 dB	0.79	0.98	13.16	12.16
15 dB	0.44	0.94	19.24	14.78
20 dB	0.72	0.97	14.72	12.18
25 dB	0.48	1.18	14.36	12.15

**Table 7 micromachines-16-01117-t007:** Important performance comparisons between different digital control attenuators.

Ref.	Size	Frequency (GHz)	Accuracy (dB)	Insertion Loss (dB)	Return Loss (dB)
[15]	2.45 × 1.75 mm^2^	DC~12	**—**	≤8.3	≤12
[19]	2.45 × 4.34 mm^2^	DC~20	≤2.22	≤1.5	≤11.95
[24]	3.2 × 3.2 mm^2^	DC~20	≤1.8	≤2	≤10
[25]	2.4 × 4 mm^2^	1~25	≤2.5	≤0.4	≤6.8
This work	0.66 × 1.38 mm^2^	DC~20	≤1.18	≤1.65	≤12.15

## Data Availability

The data that support the findings of this study are available from the corresponding author upon reasonable request.

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
