# Peer review of "Design and Preparation of Compact 3-Bit Reconfigurable RF MEMS Attenuators for Millimeter-Wave Bands"

_micromachines, 2025, doi:10.3390/mi16101117_

Round 1

Reviewer 1 Report

Comments and Suggestions for Authors

Authors present 3-bit attenuator. The manuscript has appropriate sequence to show their idea and achievement. However, revision will improve the overall quality of soundness. In many images, I cannot understand the structure, principles, or features.

  1. Move the simulation to result: The result section has insufficient information.
  2. Move fabrication and design to forward: The flow of each section is not logical, which causes discomfort to the reader.
  3. Develop all images are informative: When published, all images included in the manuscript must be visually appealing, numbers and text must be legible, and descriptions must be provided.

Reviewer 2 Report

Comments and Suggestions for Authors

The paper deals with the study of a 3-bit RF MEMS-based attenuator for the millimeter wave band. A compact structure is proposed to improve existing configurations.

After an introduction to describe the state-of-the-art for RF MEMS attenuators, a design is proposed to surpass the literature results.

The T-network attenuator integrated with a Y-shaped power divider architecture reveals that this solution is competitive or superior compared to the classical SPDT devices, more reliable, and with better power handling capabilities.

The comparison between design and experimental results is impressive; the paper is well-written, and the reviewer recommends its publication after a few corrections as indicated below.

Typos

In Table 1: The 1sh bit -> … 1st

In Table 4 and 5: Accuraacy -> Accuracy

Reference 10 is incomplete

Conflicts of Interest: The autshors declare no conflict of interest -> … authors …

Reviewer 3 Report

Comments and Suggestions for Authors

This paper designs and implements a Radio Frequency Micro-Electro-Mechanical Systems (RF MEMS) reconfigurable attenuator, which uses RF MEMS switches to switch between multiple attenuation levels.

There have been multiple literature reports on RF MEMS reconfigurable attenuators. Please elaborate on the innovation points of the RF MEMS reconfigurable attenuator proposed in this paper, its differences from existing structures, the advantages of the structure in this paper, and the advancement of its performance indicators.

RF MEMS switches are the key components for realizing this reconfigurable attenuator. However, the paper does not provide the mechanical design of the RF MEMS switches, and the test results of the 6 switches in the attenuator (including parameters such as driving voltage, switching time, and contact resistance). Please supplement the above content.

The test curves for 20 dB and 25 dB in Figure 10(a) show significant fluctuations. Please explain the reasons for this phenomenon.

Round 2

Reviewer 3 Report

Comments and Suggestions for Authors

Please provide the detailed process of the switch test results, including the test method, test equipment, test curves, etc.

Author Response

 Thank you very much for your thorough evaluation and valuable comments on our manuscript. Your insights and suggestions have been extremely helpful in improving the quality of our work. We appreciate the time and effort you have dedicated to reviewing our paper.
